# UNESCO's Contribution to Face Global Water Challenges

**Alexandros K. Makarigakis * and Blanca Elena Jimenez-Cisneros**

International Hydrological Programme (IHP), Natural Sciences Sector, UNESCO, 7 Place de Fontenoy, 75007 Paris, France; bjimenezc@iingen.unam.mx

**\*** Correspondence: a.makarigakis@unesco.org; Tel.: +33-1-456-80806

**Abstract:** The current world population of 7.6 billion is expected to reach 8.6 billion in 2030, 9.8 billion in 2050 and 11.2 billion in 210, with roughly 83 million people being added every year. The upward trend in population size along with an improved quality of life are expected to continue, and with them the demand for water. Available water for human consumption and development remains virtually the same. Additional to the different pressures of the demand side on the available resources (offer side), climate variability and change apply further pressures to the management of the resource. Additional to the increase in evaporation due to temperature rise, climate change is responsible for more frequent and intense water related extreme events, such as floods and droughts. Anthropogenic activities often result in the contamination of the few pristine water resources and exacerbate the effects of climate change. Furthermore, they are responsible for altering the state of the environment and minimizing the ecosystem services provided. Thus, the water security of countries is compromised posing harder challenges to poor countries to address it. This compromise is taking place in a complex context of scarce and shared resources. Across the world, 153 countries share rivers, lakes and aquifers, home to 40% of the world's current population. The United Nations Educational, Scientific and Cultural Organization (UNESCO) is the scientific arm of the United Nations and its International Hydrological Programme (IHP) is the main vehicle for work in water sciences at an intergovernmental level. IHP VIII, IHP's medium term strategy, aims to assist UNESCO's Member States (MS) in achieving water security by mobilizing international cooperation to improve knowledge and innovation, strengthening the science-policy interface, and facilitating education and capacity development in order to enhance water resource management and governance. Furthermore, the organization has established an Urban Water Management Programme (UWMP) aiming at promoting sustainable water resource management in urban areas.

**Keywords:** climate change; IHP; intergovernmental; science and technology; sustainability; UNESCO; water management; water security; Urban Water Management Programme

## 1. Introduction

The International Hydrological Programme (IHP) is the only intergovernmental programme of the UN system devoted to the scientific, educational, cultural and capacity building aspects of hydrology for the better management of water resources. Drawing on more than four decades of experience, UNESCO-IHP fosters and consolidates cross-disciplinary and cross sectoral networks that facilitate cooperation within research and capacity building, and development of analytical tools and data sharing, primarily across national boundaries. UNESCO-IHP also enhances awareness raising of policy-makers at the national, regional and international level on the predictions and risks related to global change, including climate change and human impact.

IHP is a truly intergovernmental programme, having its planning, definition of priorities, and supervision of the execution to be decided by the Intergovernmental Council. The Council is composed of 36 UNESCO Member States elected by the General Conference of UNESCO at its ordinary sessions held every two years. Equitable geographical distribution and appropriate rotation of the representatives of the Member States are ensured in the composition of the Council. Each of UNESCO's six electoral regions elects Member States for membership in the Council.

Consequently, the Council elects a chairperson and four vice-chairpersons. These, with the chairperson of the previous Bureau as ex-officio member, constitute the Council's Bureau. The composition of the Bureau so formed reflects an equitable geographical distribution, each representing UNESCO's six electoral regions. The members of the Bureau remain in office until a new Bureau has been elected (It needs to be noted that following the 23rd session of IHP's Intergovernmental Council, the role of the ex-officio member will no longer apply and Member States will elect a chairperson, a rapporteur and four vice-chairpersons).

Responding to the need to have an impact on the practical management of water resources, IHP networks comprise not only the scientists but also professionals, different sectors, and the society at large, including youth, gender and children groups. There is no other international Member States' water network with such a wide range of disciplines, sectors and stakeholders.

## 2. Intergovernmental Hydrological Programme: Origin and Strategy

At the end of the first International Hydrological Decade (IHD, 1965–1974) the international scientific community together with governments realized that water resources often were one of the primary limiting factors for harmonious socio-economic developments in many regions of the world. Moreover, they realized that to solve problems, internationally coordinated cooperation mechanisms were necessary to enhance the knowledge base, capacity and rational management. This gave birth to the UNESCO's IHP.

IHP facilitates an inter- and transdisciplinary integrated approach to watershed and aquifer management, incorporating the social, economic and human dimensions of water resources. To advance knowledge development and dissemination, IHP uses all available experience and promotes and develops international cooperative research in hydrological and freshwater sciences. IHP was planned and implemented in six-year phases, covering themes reflecting the current priorities decided by Member States; as of 2014, the planning exercise has shifted to an eight-year cycle.

The core themes of the first three phases of IHP (1971–1989) followed the same directions of the International Hydrological Decade, focusing on research and capacity building in hydrological science in its strict sense. Since then, the different phases of IHP (Figure 1) were always in advance of the major challenges the world had to face concerning water.

In the nineties, more than 25 years prior to the Agenda 2030 and the Sustainable Development Goals, the programme, being in its fourth phase, IHP-IV (1990–1995), identified sustainability and water resource development and management as key elements, adopting "Hydrology and Water Resources for Sustainable Development" as a core theme. Similarly, the work in the fifth phase, IHP-V (1996–2001), had "Hydrology and Water Resources Development in a Vulnerable Environment" as a core theme.

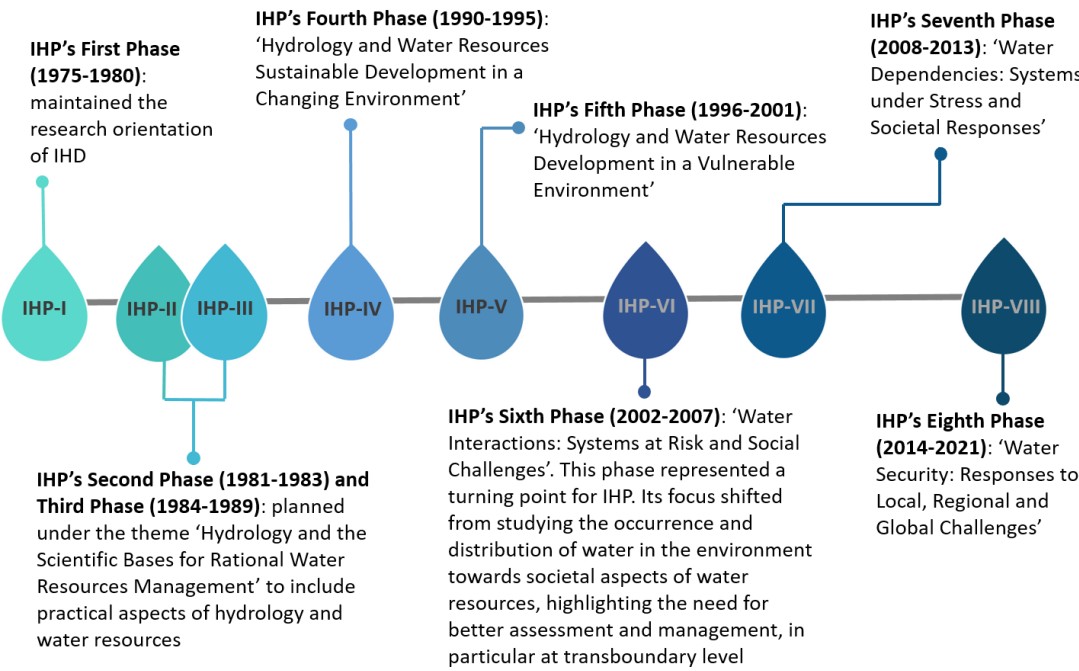

**Figure 1.** IHP Phases.

UNESCO, being the scientific arm of the UN family is required to lead the work in water breaking scientific barriers using an out of the box approach.

Recognizing the need for a paradigm shift in thinking on water from fragmented compartments of scientific inquiry to a more holistic, integrated approach, the core theme for IHP-VI (2002-2007) was defined as "Water Interactions: Systems at Risk and Social Challenges". The same trend continued in the formulation of the IHP-VII (2008–2013), which adopted 'Water Dependencies: Systems under Stress and Societal Responses' as a core theme, further emphasizing the interacting dependencies of the system components and the important role of society. All these themes were well in advance of the national research agendas setting new trends on the need to develop knowledge.

Following the 2000–2015 and the Millennium Development Goals, Member States came to an agreement for establishing an ambitious and interconnected development agenda of 17 Sustainable Development Goals (SDGs), Agenda 2030. Sustainable Development Goal 6 aims at ensuring availability and sustainable management of water and sanitation for all. A closer look at the SDGs reveals that many of the SDGs have a strong relationship with sustainable water use and consumption, e.g., SDGs 2 (zero hunger), 3 (good health and well-being), 11 (sustainable cities and communities), 12 (responsible production and consumption), 13 (climate action), and 14 (life below water).

UNESCO's focus of its eighth programmatic phase IHP VIII (2014–2021), has adopted 'Water Security: Responses to Local, Regional and Global Challenges' as its overarching idea. Given population growth, deteriorating water quality, the growing impact of floods and droughts and the other hydrological effects of global change, water security is a growing concern. It touches upon all aspects of life and requires a holistic approach, which actively integrates social, cultural and economic perspectives, scientific and technical solutions and attention to societal dynamics. In 2016, the World Economic Forum identified the water crisis as the global risk of highest concern for people and economies over the next ten years.

Water security has been defined by UNESCO's Member States as the capacity of a population to safeguard access to adequate quantities of water of an acceptable quality for sustaining human and ecosystem health on a watershed basis, and to ensure efficient protection of life and property against water-related hazards such as floods, landslides, land subsidence, and droughts. To date this is the only intergovernmentally approved definition. Although other better-rounded definitions have been developed, their use often causes political challenges and thus are frequently avoided.

The activities of the eighth phase of IHP (IHP-VIII) are being conducted along three strategic axes: (a) mobilizing international cooperation to improve knowledge and innovation to address water security challenges; (b) strengthening the science-policy interface to achieve water security at all levels; and (c) developing institutional and human capacities for water security and sustainability.

The work of IHP-VIII focusing on capacity building and awareness raising on six thematic areas to assist Member States in their challenging endeavor to better manage and secure water and to ensure the necessary human and institutional capacities. These are:

- Theme 1: Water-related Disasters and Hydrological Changes
- Theme 2: Groundwater in a Changing Environment
- Theme 3: Addressing Water Scarcity and Quality
- Theme 4: Water and Human Settlements of the Future
- Theme 5: Ecohydrology, Engineering Harmony for a Sustainable World
- Theme 6: Water Education, Key to Water Security

IHP-VIII on water security aims to address challenges identified in Agenda 2030, Sendai Framework for Disaster Risk Reduction and the New Urban Agenda and the Paris Agreement.

Within the framework of water security, IHP builds capacity of member states by synergistically integrating the experience and tools available within the activities implemented in all six thematic areas. The goal is to provide the scientific knowledge base for sound policy advice, in order to manage and cope with challenges to water resources in the practice, and to increase the resilience of natural and human systems with an emphasis on vulnerable communities.

## 3. IHP's Urban Water Management Programme (UWMP)

IHP's UWMP aims to promote sustainable water resource management in urban areas by helping countries develop and implement effective strategies and policies for urban water management through the dissemination of scientifically-sound policy guidelines, scientific knowledge and information on new and innovative approaches, solutions and tools for sustainable urban water management, as well as by providing capacity building support on key urban water issues.

## 4. Global Water Challenges

A frequent expression used by numerous professionals to describe water related challenges is that "there is too much, too little or too polluted". UNESCO, within the concept of water security, is working to ensure that all three challenges are addressed. As it is quite difficult to capture the results of the work of more than 3000 professionals comprising UNESCO's Water Family, a few selected issues are presented below.

### 4.1. Pressures on Water Availability

#### 4.1.1. Population Growth

We currently live in the Great Acceleration period of the Anthropocene. In tracking the effects of human activity upon the Earth, a number of socioeconomic and earth system parameters are utilized including population, economics, water usage, food production, transportation, technology, green-house gases, surface temperature, and natural resource usage. Since 1950, these trends are increasing significantly if not exponentially. The current world population of 7.6 billion is expected to reach 8.6 billion in 2030, 9.8 billion in 2050 and 11.2 billion in 2100 [1].

In 1990, 43% (2.3 billion) of the world's population lived in urban areas. In 2015, the urban population had grown to 54% (4 billion) and it is expected to increase to 66% by 2050. It is projected that 2.5 billion people will be added to urban populations by 2050, 90% of which will be in Asia and Africa [2,3]. The urbanization trend has experienced a remarkable increase in the absolute numbers of urban dwellers, from a yearly average of 57 million between 1990–2000 to 77 million between

2010–2015. It has to be noted that, although cities impose challenges on the environment, natural resources and the hydrological cycle, no questioning to this model for development has been made.

As the world's population has increased by around 4-fold in the 20th century, the human water consumption has increased around 5, 18 and 10 times for agricultural, industrial and municipal use, respectively [4,5]. A rise in the world population and its standards of living, along with unsustainable practices, has put water resources under ever-increasing pressure globally.

### 4.1.2. Agriculture

Agriculture is the world's largest user of water. Considering water abstraction, agricultural use of water represents near 70% of the global use [6] with clear differences among developed countries and developing countries where rain-fed irrigation accounts for 60% of their production. Irrigation water withdrawal in developing countries is expected to grow by about 14 percent from the current 2130 km$^3$ per year to 2420 km$^3$ in 2030 [7].

In addition to this use of water to produce crops, water is also used to manufacture food. In Europe, for example, the manufacturing of food products consumes on average about 5 m$^3$ of water per person per day [8]. At the same time, with as much as 1.3 billion tons of food wasted annually [9], 250 km$^3$ of water is being "lost" per year due to food waste (food waste can be defined as the discarding of food that was fit for human consumption but has become spoiled, expired or otherwise unwanted) worldwide [9]. At the global level, meat and cereals clearly stand out in the global proportion of food waste by 21.7% and 13.4%, respectively [10].

### 4.1.3. Water Scarcity

By 2050, it is estimated that 40 per cent of the global population will be living in river basins that experience severe water stress, particularly in Africa and Asia. Approximately 450 million people in 29 countries face severe water shortages [11]; about 20% more water than is now available will be needed to feed the additional three billion people by 2025; as much as two-thirds of the world population could be water-stressed by 2025 [12]; Water scarcity is projected to become a more important determinant of food scarcity than land scarcity, according to the view held by the UN [13].

### 4.1.4. Climate Variability and Change

Climate variability and change intensifies in a significant manner such water-related threats [14]. A recent model intercomparison study reveals that 2 °C of global warming will result in a severe decrease in the available water resources for 15% of the global population and will increase the number of people living under absolute water scarcity by another 40% compared to the effect of population growth alone [15]. Furthermore, numerous studies show that warming weather can trigger more water use and aggressive extraction from water resources [16–18], which together with changes in operation patterns [19–23] pose additional pressure on the already scarce water resources.

Changes in global climate are also expected to reduce groundwater recharge to aquifers, storage and discharge [24]. These reductions will have significant negative effects on available groundwater for development as well as for groundwater dependent ecosystems and the services they provide to both humans and the environment. Moreover, in the case of coastal aquifers, the combination of groundwater level drop and sea level rise due to the direct and/or indirect effects of climate change will cause an increase in saltwater intrusion, which in turn will pose serious threats to the livelihoods of one of the most vulnerable populations to climate change: islanders.

The role public and non-governmental organizations, including research and academic organisations, play in enabling adaptation at multiple scales has been shown to be crucial by recent studies [25].

### 4.2. Water Quality

Further to the pressure for additional water sources due to population growth and increase of the standard of living, human activities have increased the release of various contaminants in ground and surface water resources, resulting in a wide range of consequences from major decline in water availability and water quality to massive environmental changes. Half of the world's rivers and lakes are polluted; and major rivers, such as the Yellow river, Ganges river, and Colorado river, do not flow to the sea for much of the year because of upstream withdrawals [26]. Inefficient and ineffective use of fertilizers and pesticides in agriculture is a major contributor to ground and surface water contamination. As an example, groundwater pollution in Greece is often related to the use or abuse of fertilizers, which diffuse into soils and contaminate the aquifers. Additionally, coastal aquifers are subject to a negative water balance, owing to their overexploitation that triggers saline water intrusions [27]. Ineffective waste management often results in the release of contaminants related to noxious compounds whereas fertilized agricultural fields and wastewater treatment plants often discharge nutrients in significant quantities [28].

Such unfolding issues result in an increasing cost for water treatment and have exacerbated sociopolitical tensions over decreasing water availability, which have made water management and controlling the competition over water allocation extremely complex and sensitive [29].

Finally, water quality can also be largely degraded by climate change [6,30,31], although a comprehensive global understanding of water quality consequences of climate change is currently lacking.

There is a consensus however, that there could be significant water quality issues resulting from planned and unintended responses to climate change. Thus, any plan to address undesirable water quality impacts will require a holistic approach integrating activities of institutions responsible for managing air, land and water resources.

UNESCO [32] estimates that around 2 billion people currently live in water-stressed areas and over 800 million people have inadequate access to safe drinking water, which is supported by the findings of the latest Joint Monitoring report [33], stating that 844 million people still lacked even a basic drinking water service.

### 4.3. Water Related Hazards

Water-related hazards account for around 90% of all natural hazards globally, marking floods and droughts as the two most destructive natural threats to human societies. Climate change is expected to cause a rise in their intensity and frequency of extreme water events. Only throughout 2010, water-related disasters killed nearly 300,000 people, affected around 208 million others and cost nearly $110 billion [32,34]. Hence, recent years have seen increased attention for strategic flood risk assessments, and their inclusion in global integrated assessments [35].

In addition, the impact of extreme events on water-related hazards is expected to also become more intense and more geographically spread under climate change conditions causing an increase on social vulnerability. Recent multi-model studies highlight a likely increase in the global severity of drought by the end of the 21st century, in which the frequency of drought increases by more than 20% in highly populated regions, such as South America and Central and Western Europe [36]. Drought due to reduced rainfall has been the cause of a 95% reduction in Lake Chad's area between 1960 and 1985 [37]. Albeit water levels have risen since the 2000s, ecosystems have been significantly imbalanced and weakened, unable to provide livelihood related services. Furthermore, conflicts plagued the area over access to the diminished resource, and significant waves of migration have occurred through the years [38].

Water-related hazards are continuously present in the local and global news. While California is currently recovering from a major 5-year drought [39], southern Quebec was in a state of emergency due to major flooding in the area, including parts of Montreal [34,40]. On 18 January 2018, Cape Town residents, South Africa's second-largest city, woke up to their Mayor's, Ms Patricia De Lille,

proclamation that "Day Zero," the day the city would run out of water, was fast approaching. On February 1—the height of summer in the southern hemisphere, when water demand is greatest—the city clamped down, harder than any city in the world with its living standards. Officials set a target of 50 Liters (13 gallons) per person, per day, for all domestic uses: cooking, bathing, toilet flushing, washing clothes. Watering lawns and scrubbing cars with city water had already been banned for months. "The abuse of water means that we will all suffer," De Lille had warned.

On the other hand, an increase in flooding frequency is projected in more than half of the world, particularly in the non-snow dominated regions, which naturally have a greater population [41]. The recent flood in Houston, Texas, during the course of hurricane Harvey, resulted in more than 80 fatalities and an estimated economic cost that exceeded $150 billion USD [42]. Globally, economic losses from flooding exceeded $19 billion in 2012 [21], and have risen over the past half century [30,43]. Hence, recent years have seen increased attention for strategic flood risk assessments, and their inclusion in global integrated assessments [35].

Only throughout 2010, water-related disasters killed nearly 300,000 people, affected around 208 million others and cost nearly $110 billion [32]. Investing in disaster risk reduction is thus a precondition for developing sustainably in a changing climate. It is a precondition that can be achieved and that makes good financial sense.

*4.4. Servicing the Most Vulnerable: Slum Populations*

Even though significant progress has been made globally towards improving access to water, almost 700 million people still lack access to clean drinking water globally [43]. Informal settlements (slums) constitute a significant percentage of the urban population. There were more slum dwellers in 2012 than in 2000, a trend that will likely continue in the future [44].

Slum dwellers most often lack water and sanitation related services, as well as many other public services. For instance, in India, 56% of the population in the top 20% (household income group) has access to piped water, compared to 6% of the bottom 20% [32]. Furthermore, they often have to pay higher rates to receive water than citizens covered by the piped network do; water price can be up to thirty three times higher than the one charged by the operators [45]. Sanitation facilities are usually non-existent, having people frequently relying on communal toilets or open defecation.

Slum populations constitute the most vulnerable in an urban or peri-urban setting, having high exposure to natural hazards, often settled in areas that are not in the city plan and which are not suitable for human settlements, such as flood plains or hill sides prone to landslides. Rapid urban expansion aggravates these challenges and the people are also disproportionately affected by the impacts of climate hazards [2,33].

## 5. UNESCO's Contribution to Global Water Challenges

UNESCO is the scientific organization of the United Nations, whose purpose is advancing, through the educational, scientific, and cultural relations of the peoples of the world and the free exchange of ideas and knowledge, the objectives of international peace and of the common welfare of mankind.

Addressing the increasing global water challenges will be achieved by narrowing the problem within the broader framework, including the UNESCO mandate to go beyond the Integrated Water Resources Management (IWRM), water-energy-food-environment nexus or new holistic frameworks. These frameworks should aim to open up a dialogue with practitioning, management and policy communities.

As presented in Figure 2, IHP works on the basis of three axes to address global water challenges: networking, science-policy interface and building/strengthening both institutional and human capacity. In the text below, a few examples of the type of activities being implemented is provided.

Facing these global challenges requires pushing the boundary of current advancements within the water security domain. First, a common acceptance and international recognition of the water

security concept and its political acceptance should be made to facilitate progress in this field. Secondly, new insights, tools and methodologies are needed for better representation of complex interactions within coupled human and natural systems, especially in urban regions across a range of temporal and spatial scales. Such attempts should be made with the greater goal of diagnosing water-related threats as a result of extreme or gradual changes in natural and anthropogenic conditions, in light of current limitations in future projections [46]. Thirdly, an entire change in the general mindset society has towards water and water related issues is needed to effectively minimize the increasing challenges and to eliminate new ones. At a later stage, scientific knowledge needs to be acquired and used to change cultural aspects through education.

Scientific contributions, addressing the above-mentioned challenges, are emerging [46]; however much more needs to be done. First, new technologies are required to implement the scientific solutions, particularly with respect to water conservation, treatment, and reuse. While there are practicable water conservation technologies around, much more is needed in the water quality domain, particularly with respect to operational regulation and exotic contaminants.

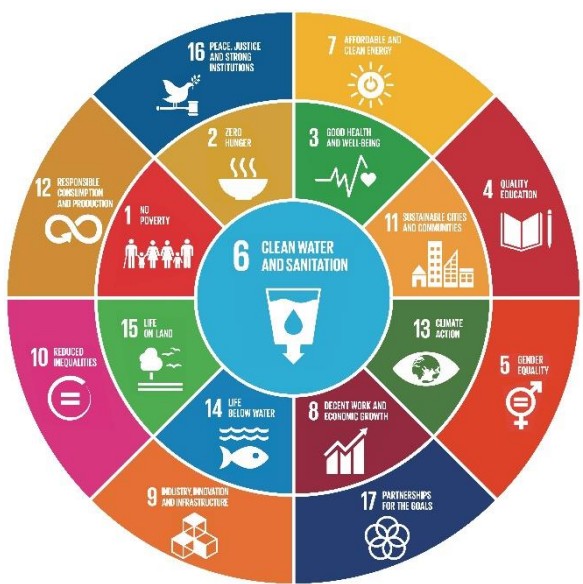

**Figure 2.** The water centric 17 Sustainable Development Goals.

*5.1. Science & Technology (Tools and Methodologies)*

5.1.1. Water Availability

IHP's "Water for Human Settlements"and its UWMP, have been focusing their efforts in disseminating and promoting the use of artificial intelligence (AI) and of the internet of things (IoT) in urban water management to address both the issue of water availability in an urban setting, as well as water quality. The use of sensors, transmitters, Supervisory Control and Data Acquisition (SCADA) systems, modelling and other tools, can effectively reduce the non-revenue water (NRW) into single digits and ensure enough water of good quality for people. Furthermore, the Programme and its urban based initiative (UWMP) have been focusing on intermittent water supply in order to better understand how this could be avoided or done in a secure and safe manner, to the extent possible.

5.1.2. Water Quality

Water quality can be remotely monitored via the use of satellite information. The International Initiative on Water Quality [47] has initiated a project that supports monitoring of Sustainable Development Goal 6's targets 6.3.2 (Proportion of bodies of water with good ambient water quality) and 6.6.1 (Change in the extent of water-related ecosystems over time), where remote sensing technology

is used to provide a time series of information related to the pollution (or absence) of the surface water bodies. The UWMP is currently studying the effects of flooding episodes on water quality and a publication to this extent is expected in late 2019.

### 5.1.3. Water Related Hazards

In order to enhance the resilience of communities to floods and droughts, IHP has been developing the Flood and Drought Monitor [48] and has been providing the technology to various regions around the globe. The monitor permits the forecasting of extreme events (flooding, drought) well before they take place. In order to provide an end to end solution, similar tools have been developed, such as the drought atlas [49], which is coupled with a forecasting system and telephone apps that provide information to farmers in Latin American countries related to their crops' irrigation.

### 5.1.4. Nature Based Solutions: Ecohydrology

The fifth annual theme-oriented report of the United Nations World Development Report (WWDR) produced by UNESCO's World Water Assessment Programme focuses on opportunities to harness the natural processes that regulate various elements of the water cycle, which have become collectively known as nature-based solutions (NBS) for water [49]. IHP's work on ecohydrology [50] promotes the use of the interactions between biota and hydrology to regulate, remediate and conserve ecosystems to stabilize and improve the quality of water resources. Implementation of ecohydrology is undertaken through "harmonization" with existing and planned hydrotechnical infrastructures. Twenty-three pilot projects have been established worldwide to validate and quantify the effectiveness of ecohydrological solutions [50]. It needs to be stressed that the application of ecohydrology principles can be utilized to provide clean potable water, as well as to minimize the effects of water related hazards to communities and the environment.

UWMP has produced knowledge on this topic focused on an urban setting with two publications: "Capacity building for ecological sanitation: concepts for ecologically sustainable sanitation in formal and continuing education" and "Aquatic Habitats in Sustainable Urban Water Management".

### 5.1.5. Data Management

Data need to be stored in a safe environment that allows its analysis with a view to produce information for improved water resource management and decision making. Various databases and platforms have been developed during the past decades, usually at an inhibitory cost to developing states; especially when multiple licenses are required. IHP's Water Information Network System (WINS) is an open access, open source platform for sharing, accessing and visualizing water-related information, as well as for connecting water stakeholders [51]. WINS allows access to various types of information (maps, reports, graphs, etc.) covering the entire water cycle, ranging from groundwater to urban water through gender issues, from a local to a global scale. Information provided in the form of maps can be combined directly on the platform in order to create new information, and generate customized maps that can be shared with a large panel of stakeholders such as policy makers, institutions, researchers, or the civil society.

### 5.2. International Cooperation

Transboundary basins cover more than half of the Earth's land surface, account for an estimated 60% of global freshwater flow and are home to more than 40% of the world's population. Across the world, 153 countries share rivers, lakes and aquifers, and 592 transboundary aquifers have been inventoried by UNESCO's International Hydrological Programme to date. Transboundary water cooperation is thus critical for ensuring sustainable management of water resources.

IHP's PCCP (From Potential Conflict to Cooperation Potential) project facilitates multi-level and interdisciplinary dialogues in order to foster peace, cooperation and development related to the management of transboundary water resources. The project follows the idea that although

transboundary water resources can be a source of conflict their joint management can be strengthened and even used as a means for further cooperation, contributing to UNESCO's mandate: to nurture the idea of peace in human minds.

Further to initiatives such as PCCP, UNESCO and UNECE are co-custodians of the SDG 6 indicator 6.5.2 on water cooperation, who provided in July 2018 the first global baseline. The work to date reveals that although significant progress has been made, arrangements for transboundary water cooperation are often absent.

UNESCO was designated as the agency to lead the United Nations International Year of Water Cooperation (IYWC) in 2013. The organization mobilised an estimated 25 million people around the world that year, positioning the idea to cooperate instead of compete/fight among countries, regions and different stakeholders to manage water.

At an urban level, UNESCO via its publications has examined and analysed urban water conflicts, their origins and nature, and have presented several historical urban water conflict cases and illustrations of changing conflict nature, including a theoretical analysis of ecological–economic factors to provide a basis for urban water conflict solution guidelines [52].

*5.3. Science-Policy Interface*

A collaborative, two-way interaction between science and policy spheres is the key to achieving practicable water security solutions.

As an intergovernmental organization, UNESCO's efforts are mainly focusing on decision makers. Tools developed are designed to be simple to use and contain the information that is required to make a science-based decision.

Recently, IHP established a Science Policy Interface Colloquium in Water (SPIC Water) as part of its Water Dialogues framework. The 1st SPIC Water took place on 14 June 2018 at UNESCO's Headquarters in Paris, France, and brought together ministers responsible for water resource management in 13 countries, along with experts and representatives of Member States [53]. The Colloquium was an opportunity to take stock of the progress made towards achieving the Sustainable Development Goal on Water and Sanitation (SDG6). It was organized at the request of Member States to discuss how UNESCO's International Hydrological Programme (IHP) can help to identify science-based solutions, effective policies and practices on water and sanitation, and support countries in their efforts to implement the 2030 Agenda.

The ministerial messages highlighted that the 2030 Agenda is promoting local action and positive changes in institutions at the country level. However, the sustainability of actions remains a challenge. They also noted the need to harmonize activities and policies at the global, regional and local level and to adapt targets to the local context. All underlined the need for reinforced human capacity if the 2030 Agenda was to be implemented in the domain of water. They welcomed the existence of a forum like SPIC Water, where policy-makers could exchange viewpoints with experts, who provide the knowledge and information needed to adapt policies based on available knowledge.

The Science Policy Interface Colloquiums on Water will play a significant role in the implementation of the 2030 Agenda. SDG 6 provides the platform for decision makers at the highest political level in water resource management to express the challenges they face and for scientists to adjust their work to cater for their needs. It will thus, guide future research and scientific work to pursue solutions that can be applied by countries.

SPIC water is designed to complement existing international fora, such as the World Water Forum, Dushanbe Conference, International Water Weeks, etc., and feed into the discussions during the High Level Political Forum in New York, when SDG 6 is examined.

### 5.4. Human Capital

The uptake of scientific and technological solutions requires particular attention to the socio-economic drivers at the managerial and public levels. The importance of social capital cannot and should not be underestimated in achieving water security.

The availability of sound scientific and technological tools cannot provide a solution to water resource management alone; it requires trained professionals to use them, sensitized decision makers to understand their importance and informed citizens to accept their results. Capacitating the human capital is thus the main focus of UNESCO IHP's investment.

An average of 10,000 experts, decision makers and communities have been trained and/or been made aware of various issues related to water security over the past two years (2016–2017) in a wide range of themes (Figure 3) by the efforts of UNESCO's Water Family.

Training on issues of water security in an urban context have been spearheaded by UNESCO's water related Chairs and Category 2 Centres.

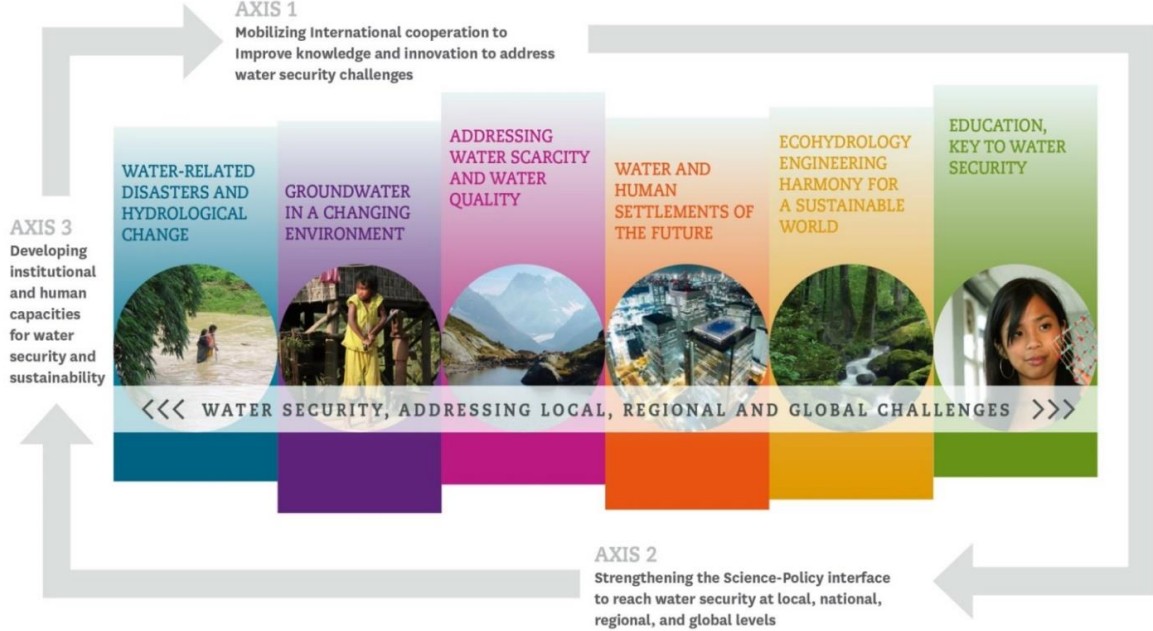

**Figure 3.** IHP VIII (2014–2021); Water Security: Responses to Local, Regional and Global Challenges.

### 5.5. Networking

UNESCO's Water Family is a network of networks comprising of 169 National IHP Committees or focal points, 37 Category 2 Centres and 50 UNESCO Chairs, along with the Secretariat of the UNESCO Water Science Division, IHP, WWAP and regional offices that surpass in numbers the staggering amount of 3000 experts.

Working quite often within the framework of IHP's International Initiatives [54] and/or within the implementation of projects and activities in the framework of IHP VIII, UNESCO's Water Family provides technical support to Member States in achieving water security and through this, internationally agreed goals, such as SDGs 6, 11 and 13, and agreement such as the Sendai Framework, Paris Agreement and the New Urban Agenda.

The framework of IHP's 17 international initiatives (see Table 1), provides yet an additional network of experts, who do not necessarily belong to UNESCO's Water Family institutions but who contribute to the strategic goals of the organization.

**Table 1.** IHP's 17 international initiatives.

| Initiative | Description | Contact Officer |
|---|---|---|
| | Flow Regimes from International Experimental and Network Data, an international research initiative that helps to set up regional networks for analyzing hydrological data through the exchange of data, knowledge and techniques at the regional level | Mr Abou Amani a.amani@unesco.org |
| | Global Network on Water and Development Information in Arid Lands, a global network on water resources management in arid and semi-arid zones whose primary aim is to build an effective global community to promote international and regional cooperation in the arid and semiarid areas | Mr Anil Mishra a.mishra@unesco.org |
| GLOBAL NETWORK OF WATER MUSEUMS | Global Network of Water Museums, is an IHP initiative to create synergies within UNESCO with the aim of better using water museums to improve water management via communication and educational activities | Mr Alexander Otte a.otte@unesco.org |
| | Groundwater Resources Assessment under the Pressures of Humanity and Climate Change, a UNESCO-led project seeking to improve our understanding of how groundwater interacts within the global water cycle, how it supports human activity and ecosystems, and how it responds to the complex dual pressures of human activity and climate change | Ms. Alice Aureli a.aureli@unesco.org |
| | Hydrology for the Environment, Life and Policy, a new approach to integrated catchment management by building a framework for water law and policy experts, water resource managers and water scientists to work together on water-related problems | Mr Abou Amani a.amani@unesco.org |
| | Integrated Water Resources Management, an initiative implementing IWRM at the river basin level as an essential element to managing water resources more sustainably, leading to long-term social, economic and environmental benefits | Mr Alexandros Makarigakis a.makarigakis@unesco.org |
| | International Drought Initiative, an initiative aiming at providing a platform for networking and dissemination of knowledge and information between international entities that are actively working on droughts | Mr Abou Amani a.amani@unesco.org |
| | International Flood Initiative, an interagency initiative promoting an integrated approach to flood management which takes advantage of the benefits of floods and the use of flood plains, while reducing social, environmental and economic risks. Partners include the World Metereological Organization (WMO), the United Nations University (UNU), the International Association of Hydrological Sciences (IAHS) and the International Strategy for Disaster Reduction (ISDR). | Mr Abou Amani a.amani@unesco.org |
| | International Initiative on Water Quality, an international platform to strengthen knowledge, research and policy, and develop innovative approaches to tackle water quality challenges | Ms. Sarantuyaa Zandaryaa s.zandaryaa@unesco.org |

**Table 1.** *Cont.*

| Initiative | Description | Contact Officer |
|---|---|---|
|  International Sediment Initiative | International Sediment Initiative, an initiative to assess erosion and sediment transport to marine, lake or reservoir environments aimed at the creation of a holistic approach for the remediation and conservation of surface waters, closely linking science with policy and management need | Mr Anil Mishra a.mishra@unesco.org |
|  isarm | Internationally Shared Aquifer Resources Management, an initiative to set up a network of specialists and experts to compile a world inventory of transboundary aquifers and to develop wise practices and guidance tools concerning shared groundwater resources management | Ms. Alice Aureli a.aureli@unesco.org |
| LAND SUBSIDENCE INTERNATIONAL INITIATIVE | Land Subsidence International Initiative is a global IHP platform for scientific researchers and institutions, aimed at voluntarily creating the knowledge base to build, facilitate and foster cooperation concerning planning, hydrogeological sciences and water security in urban and coastal areas, by exchanging expertise and good practices for a better transfer of knowledge to public policies; | Ms. Alice Aureli a.aureli@unesco.org |
| **MAR** | Managing Aquifer Recharge, an initiative that aims to expand water resources and improve water quality with the adoption of improved practices for management of aquifer recharge (storage and recovery) | Ms. Alice Aureli a.aureli@unesco.org |
| P C → C P | From Potential Conflict to Cooperation Potential, a project facilitating multi-level and interdisciplinary dialogues in order to foster peace, cooperation and development related to the management of shared water resources | Ms. Renee Gift r.gift@unesco.org |
| **UWMP** | Urban Water Management Programme, an initiative that generates approaches, tools and guidelines which will allow cities to improve their knowledge, as well as analysis of the urban water situation to draw up more effective urban water management strategies | Mr Alexandros Makarigakis a.makarigakis@unesco.org |
| **WHYMAP** | World Hydrogeological Map, an initiative to collect, collate and visualize hydrogeological information at the global scale to convey groundwater-related information in a way appropriate for global discussion on water issues | Ms. Alice Aureli a.aureli@unesco.org |
|  | World Large Rivers Initiative, while excluding operational management, aims to establish a purely scientific global platform of researchers and institutions to develop, on a voluntary basis, the scientific foundation for integrated river research by exchanging expertise and good practice | Mr Abou Amani a.amani@unesco.org |

## 5.6. More than Science & Technology

The role of social and cultural processes in water security, and social processes ultimately should be embedded in IWRM models and the nexus as new algorithms. This can lead into new understanding of the complex dynamics between human and natural systems and can pave the way to extending the scope of risk management.

UNESCO's multidisciplinary mandate allows the organization to bring solutions that include various social and human elements in tandem with scientific and technological opportunities. It ensures that the education element stands on the top of the agenda and that cultural beliefs and customs are taken into consideration when one designs a training or a way to manage the valuable resource.

Principles of ethics in the use of technology need to be examined ensuring that services will be provided in an inclusive manner.

## 6. Conclusion

The importance of water has, at last, been receiving considerable attention at various fora (e.g., Davos World Economic Forum [55] and has been identified as an important element for development in the post 2015 agenda, receiving the sixth goal of the 2030 Agenda and having a fundamental role in the Sendai Framework and the New Urban Agenda.

UNESCO's International Hydrological Programme's role is to raise awareness of communities and decision makers alike on the importance of water in human development and the environment; to do so in an inclusive and culturally sensitive manner and to assure that a critical mass of experts exist with geographical and gender balance to support activities and policies geared towards the solution of the identified challenges.

Water resources management and service delivery face multiple challenges at the local, regional and global level. When sustainable development is thought of in combination with the conservation of the environment and the protection of people and from water related natural hazards, the principle of water security is formed.

UNESCO's International Hydrological Programme has water security as the core of its work during its current medium term strategy, IHP VIII (2014–2021) and supports Member States in their efforts to achieve it. Within this framework, IHP is developing scientific and technological tools for science based decision making, promotes international cooperation through networking, enhances the science policy interface and focuses its efforts in the education and training of the human capital at local, regional and global levels.

Furthermore, the programme operating as the scientific arm of the United Nations on issues related to Water, plays a forecasting role to ensure the identification of future challenges and that enough scientific research will be conducted towards the provision of solutions to these challenges, as they will be the center of development in the near future.

In an urban context, UNESCO's IHP-VIII fourth theme is dedicated to water for human settlements and together with the initiative on Urban Water Management they provide a platform where new technologies, methodologies and techniques can be identified and tested to achieve a holistic way of managing water resources and providing sustainable services in the face of water scarcity and other pressures (such as climate change, pollution and population growth).

An open call is made through this paper for the scientific community active in water to participate actively in UNESCO's water related activity and to become a member of the UNESCO Water Family.

**Author Contributions:** Conceptualization and methodology, A.K.M. and B.E.J.-C.; writing—original draft preparation, A.K.M. and B.E.J.-C; writing—review and editing, A.K.M.

**Funding:** The APC was funded by UNESCO's Natural Science Sector, Water Science Division.

**Conflicts of Interest:** The authors declare no conflict of interest. The funders had no role in the design of the study; in the collection, analyses, or interpretation of data; in the writing of the manuscript, or in the decision to publish the results.

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
