# Peer review of "UNESCO’s Contribution to Face Global Water Challenges"

_water, doi:10.3390/w11020388_

Round 1

Reviewer 1 Report

This article reviews the current water resource challenges and describe the contribution of the International Hydrological Program in addressing these challenges. It is well written and organized. There are some minor issues that need to be resolved.

1.      Not consistent reference citation formats. Some are numbered, while some are not. I would also suggest citing the on-line news (those marked with blue text) in the format of regular reference (i.e., numbered).

2.      There are some repetitive sentences. E.g., Line 261, Line 371

3.      L223: “though” to “thought”

4.      L245-246: need a reference here.

5.      L267-269: the transition of this sentence shall be improved.

6.      L259-264: Including some more example of the flooding events in recent years could be beneficial, such as the massive flooding in Houston, Texas during hurricane Harvey, which alone has cost more than 150$ billions of economic cost and more than 80 fatalities (Emanuel, 2017-PNAS; Balaguru et al., 2018-GRL; Du et al., 2019-STOTEN).

7.      L263: what is the meaning of 6 after “assessment” here.

8.      L283: This sentence shall be revised. I think it is the people instead of sanitation facilities “rely frequently on communal toilet”.

Author Response

Thank you very much for your proposed edits. I have changed the citation format

1.      Not consistent reference citation formats. Some are numbered, while some are not. I would also suggest citing the on-line news (those marked with blue text) in the format of regular reference (i.e., numbered).

2.      There are some repetitive sentences. E.g., Line 261, Line 371; deleted repetitive sentences on line 261; line 371 was repeated in the abstract, so it was left as is.

3.      L223: “though” to “thought”

I did not change the proposed edit on line 223; the para on line 223 brings a different view to the para 220 that states that "a comprehensive global understanding of water quality consequences of climate change is currently lacking." 

So in order to express the contrast I used :however": "There is a consensus however, though that there could be significant water quality issues resulting from planned and unintended responses to climate change."

I hope this suffices

4.      L245-246: need a reference here. A reference was added

5.      L267-269: the transition of this sentence shall be improved. Moved a paragraph to try and improve the flow of the document

6.      L259-264: Including some more example of the flooding events in recent years could be beneficial, such as the massive flooding in Houston, Texas during hurricane Harvey, which alone has cost more than 150$ billions of economic cost and more than 80 fatalities (Emanuel, 2017-PNAS; Balaguru et al., 2018-GRL; Du et al., 2019-STOTEN). Reference was added

7.      L263: what is the meaning of 6 after “assessment” here. "6" was deleted. It was a leftover reference

8.      L283: This sentence shall be revised. I think it is the people instead of sanitation facilities “rely frequently on communal toilet”. Revised to reflect edit

Reviewer 2 Report

General statement: The contribution is rightly included in the Perspective category and represents UNESCO's directions and challenges for tackling global water problems.

Strengths: A well described origin and strategy of an intergovernmental hydrological program, goals, topics and resources, which can be motivational and inspiring for other potential researchers.
Weaknesses: since this is not a typical professional article, there are formal shortcomings, such as the unlisted telephone number of the contact person; footnotes (some quite useless, eg 7). Figure 2 is a non-readable and especially text-based live link to larger lines (e.g., lines 167, 250); some of the underlined years (line 262) and once in the text, and not the end of the referenced ISBN publication (line 282). Incomplete statement in paragraph 4.4 (427-429).

Author Response

Weaknesses: since this is not a typical professional article, there are formal shortcomings, such as the unlisted telephone number of the contact person; Number was added

footnotes (some quite useless, eg 7). removed

Figure 2 is a non-readable I have sent the figure in hogh resolution; it should read OK now

 especially text-based live link to larger lines (e.g., lines 167, 250); Could be the choice of fonts? 

some of the underlined years (line 262) removed

and once in the text, and not the end of the referenced ISBN publication (line 282). edited

Incomplete statement in paragraph 4.4 (427-429). tried to improve the text 

Reviewer 3 Report

An concise and informative paper on the approach and contributions of UNESCO to global water challenges. I have no real issues with this paper however more detail on case studies (such as the S.Afr example) would be useful. For example, what can/did UNESCO contribute to water security apart from water restrictions? Does UNESCO provide advice to countries based on the World Dams Commission outcomes or are any alternatives recommended?

Author Response

For example, what can/did UNESCO contribute to water security apart from water restrictions? We do not promote water restrictions. For the case of Cape Town, we had already announced in 2007 the opportunity to utilize groundwater as an alternative resource, if managed properly. Unfortunately, not all scientific advice is taken up by decision makers.

Does UNESCO provide advice to countries based on the World Dams Commission outcomes or are any alternatives recommended?

UNESCO, as an intergovermental organization, provides advice to countries upon their request and based on its expertise that originates from the UNESCO Water Family. Indepedently. In the case of Cape Town, no advice was requested.

Reviewer 4 Report

I've just got a message, that there is enough reviews of submitte paper, so just a short crucial suggestion:

Paper was sent for the publishing in the section Urban water management in the special issue The Challenges of Water Management and Governance in Cities, but paper is not specifically dealing with the cities, nor with the urbanised areas. 

From my point of view the paper should more focus on the objectives, activities and results of the Urban Water Management Programme (UWMP), where both authors are the main persons. Without this is the paper just a general list of the UNESCO’s activities and challenges in the field of water management and hydrology.

Author Response

Paper was sent for the publishing in the section Urban water management in the special issue The Challenges of Water Management and Governance in Cities, but paper is not specifically dealing with the cities, nor with the urbanised areas. 

From my point of view the paper should more focus on the objectives, activities and results of the Urban Water Management Programme (UWMP), where both authors are the main persons. Without this is the paper just a general list of the UNESCO’s activities and challenges in the field of water management and hydrology.

Responces: As we are in a process of transforming the UWMP to reflect better UNESCO's mandate and remove elements that could be covered by other sister agencies, we decided to go more for a general picture on elements that are being addressed by the Water for Human Settlements of the Future thematic area. 

This is a sensitive matter for us, as the  Programme could be totally overhauled and I wouldn't want the latest  publication we have to make reference on a Programme that could be  eliminated / changed dramatically soon.

Round 2

Reviewer 4 Report

Formal error: lines 259-260. There is perhaps an author's comment, which should be deleted (text "259-264: Including some more example of the flooding events 259 in recent years could be beneficial, such as")

I understand the author's concerns about the UWMP, but my opinion is - when submitting the paper to a special issue The Challenges of Water Management and Governance in Cities, the paper should be focused more to this topic.

Author Response

Formal error: lines 259-260. There is perhaps an author's comment, which should be deleted (text "259-264: Including some more example of the flooding events 259 in recent years could be beneficial, such as")

You are correct but the comment is left to showcase to the other reviewers that their comments have been addressed

I understand the author's concerns about the UWMP, but my opinion is - when submitting the paper to a special issue The Challenges of Water Management and Governance in Cities, the paper should be focused more to this topic.

I added some elements on UWMP (see attached)
